# The electronic structure of benzene from a tiling of the correlated 126-dimensional wavefunction

Yu Liu[1], Phil Kilby[2], Terry J. Frankcombe[3] & Timothy W. Schmidt [1✉]

The electronic structure of benzene is a battleground for competing viewpoints of electronic structure, with valence bond theory localising electrons within superimposed resonance structures, and molecular orbital theory describing delocalised electrons. But, the interpretation of electronic structure in terms of orbitals ignores that the wavefunction is antisymmetric upon interchange of like-spins. Furthermore, molecular orbitals do not provide an intuitive description of electron correlation. Here we show that the 126-dimensional electronic wavefunction of benzene can be partitioned into tiles related by permutation of like-spins. Employing correlated wavefunctions, these tiles are projected onto the three dimensions of each electron to reveal the superposition of Kekulé structures. But, opposing spins favour the occupancy of alternate Kekulé structures. This result succinctly describes the principal effect of electron correlation in benzene and underlines that electrons will not be spatially paired when it is energetically advantageous to avoid one another.

[1] ARC Centre of Excellence in Exciton Science, School of Chemistry, UNSW Sydney, Sydney, NSW, Australia. [2] Data 61, Locked Bag 8001, Canberra, ACT, Australia. [3] School of Science, UNSW Canberra, Canberra, ACT, Australia. ✉email: timothy.schmidt@unsw.edu.au

The six-membered aromatic ring is ubiquitous in biology. It is a component of DNA and proteins, as well as woody biomass and petroleum[1]. The aromatic ring has been observed in interstellar space[2,3], and aromatic structures, in general, are thought to pervade the interstellar medium[4,5]. The conjugated six-membered ring is also the building block of graphene[6], a material with astonishing electronic properties.

Benzene is the archetypal aromatic molecule, displaying exceptional chemical stability as compared with other unsaturated hydrocarbons. It was first reported in 1825 by Faraday[7], and its electronic structure has been discussed for over a century. Many structural formulae have been proposed, with the most satisfying single structure being that with alternating single and double bonds, proposed by Kekulé[8,9]. However, this structure does not capture the sixfold symmetry required to satisfy experimental evidence. To account for this, Kekulé proposed an oscillation between these structures, despite knowing nothing of electrons or quantum mechanics.

The discovery of the electron in 1897 led Lewis to propose the two-electron chemical bond in 1902, although this idea was not published until 1916[10]. Following the introduction of quantum mechanics, Pauling showed that both Kekulé structures, which have the same energy, would resonate to bring about a lower energy form[11]. Hückel advanced a different approach, in which electrons would occupy molecular orbitals (MOs), delocalised over multiple atomic centres[12].

As chemical electronic structure theory advanced in the ensuing decades, there emerged a competition between the MO approach, which offers computational simplicity, and the valence bond approach, which accords with, and engenders chemical intuition[13]. It has been suggested that banana (bent) double bonds are a superior description to the $\sigma - \pi$ structure which is usually offered in chemical textbooks[14], and this has been taken as evidence that the Kekulé description is more authentic than the delocalised, MO picture[15–18]. Empedocles and Linnett went further, and suggested a decoupling of the two spin sets of electrons such that the two sets of electron spins occupy alternate Kekulé structures[19,20]. Nevertheless, the delocalised canonical MOs of benzene are usually invoked to explain spectroscopic phenomena.

It has long been known that, after sufficient computational effort, the MO and VB approaches converge to the same description of electronic structure, and are thus equally valid as computational formalisms[21]. The MO approach has won out, on the whole, for reasons of computational simplicity, and it has been argued that this is at the expense of chemical insight[13].

But, interpreting the electronic structure of a molecule in terms of the one-electron spin orbitals generated by MO theory has several major and irreparable flaws. Since the seminal work of Slater[22], it has been known that a spatial wavefunction must exhibit anti-symmetry with respect to the exchange of like-spin electrons: The sign of the spatial part of the wavefunction must change when two electrons of the same spin swap positions. This fundamental property of half-integer spin particles is the origin of the Pauli exclusion principle, and led to the development of Slater determinants to describe simple wavefunctions. The MOs are non-unique with respect to the anti-symmetrised wavefunction since the determinant is invariant upon any unitary transform of the occupied orbitals. Furthermore, even if one were to insist upon employing the canonical MOs which result from correctly anti-symmetrised Hartree–Fock theory, interpretations that view these orbitals individually ignore anti-symmetry: The Hartree product is not a correct wavefunction[22].

However, we may inspect the $3N$-dimensional electronic wavefunction ($3N = 126$ in the case of benzene) that results from any theoretical framework (including MO theory) to regain chemical insight. This should be done without imposing

(possibly erroneous) chemical intuition on the problem at hand. The electrons being described by the electronic wavefunction are fundamentally indistinguishable particles. That means that in the $3N$-dimensional space of the wavefunction there are regions that are equivalent, related through the permutation of electrons. Because these regions are equivalent and together span the space of the wavefunction, they are analogous to tiles. All of the information of the wavefunction is contained in just one of the permutable tiles[23,24].

Recently, we reported a method to identify and visualise wavefunction tiles. The procedure, known as dynamic Voronoi Metropolis sampling (DVMS), uses an iterative algorithm to identify regions in $3N$-dimensional space delineated by nodes, in the case of a boundary between differently signed tiles, and a Voronoi diagram in the case of like-signed tiles[25–29]. A Voronoi diagram is a partitioning of space into regions closest to a particular site (of several). Here, the Voronoi decomposition of the $3N$-dimensional space of the wavefunction is according to $N_\alpha!N_\beta!$ sites generated by permuting the labels on the electrons of the same spin, where $N_\alpha$ and $N_\beta$ are respectively the number of $\alpha$ and $\beta$ electrons. The DVMS algorithm is designed to find a self-consistent position $\bar{\mathbf{x}}$ for one permutable site (and thus of all the permutations defining the Voronoi decomposition) that is the centroid of a tile $R$, such that $\bar{\mathbf{x}} = \int_R \mathbf{x}\Psi^2 d\mathbf{x}$. Self-consistency is required as $\bar{\mathbf{x}}$ depends on $R$, while simultaneously $R$ is defined by $\bar{\mathbf{x}}$.

The DVMS procedure reproduces motifs such as core electrons, single bonds, multiple bonds (of the banana type) and lone pairs[25]. The method takes as its sole input the correctly anti-symmetrised wavefunction, and is thus agnostic to the theoretical approach. As such, multiconfigurational wavefunctions can be investigated to inspect the effects of electron correlation: A 2-configuration wavefunction for $C_2$ was found to exhibit a triple bond and singlet-coupled biradical electrons occupying positions on the ends of the molecule, in accord with Shaik et al.'s quadruple bond[25,30]. The DVMS procedure has also been shown to reproduce the curly arrow electron movements along a chemical reaction path[26], and connect electronic spectroscopy with the concept of vibrational normal modes[27].

In this work, we revisit the electronic structure of benzene using the DVMS procedure. It is found that a single-determinant wavefunction reproduces Kekulé structures, but with the two spins sets equivocal to the other's occupation of one Kekulé structure or the other. As the number of excited configurations is increased, the wavefunction evolves to distinctly prefer a relative disposition of the spins such that their Kekulé structures are anti-correlated.

## Results

**Kekulé structures from wavefunction tiles**. As creatures who inhabit three spatial dimensions, we find it difficult to picture objects of higher dimensionality than this. A $3N$-dimensional wavefunction tile is thus a formidable object to visualise. The tile is defined by the Voronoi site, a $3N$-vector which can be projected onto the dimension of each electron to yield a structure in real space. The Voronoi site, obtained by the DVMS procedure, for the single-determinant RHF/6-31G(d) wavefunction of benzene is shown in Fig. 1a. The electrons of each spin alternately singly or doubly bond between carbons, in accord with the Kekulé structure (s). To indicate the extent of the wavefunction tile, it can be cross-sectioned: An isosurface of the wavefunction is mapped out in the dimensions of each electron, in turn, holding the others at the Voronoi site. This results in the lobes depicted in Fig. 1b. Each blue lobe indicates the range of each C–C bonding electron in the tile, with C–H bonding electrons shown in grey. The isosurface value is chosen aesthetically[26]. Note that these sites are not unique. Starting the DVMS walk from a different point can lead to the alternate

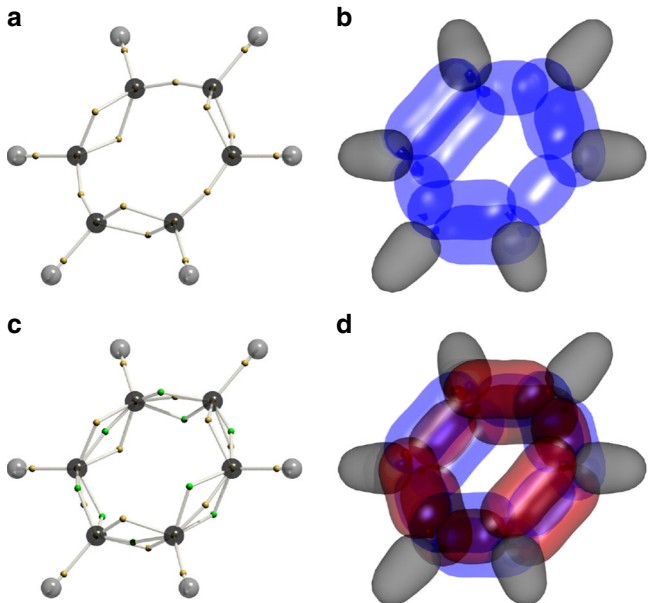

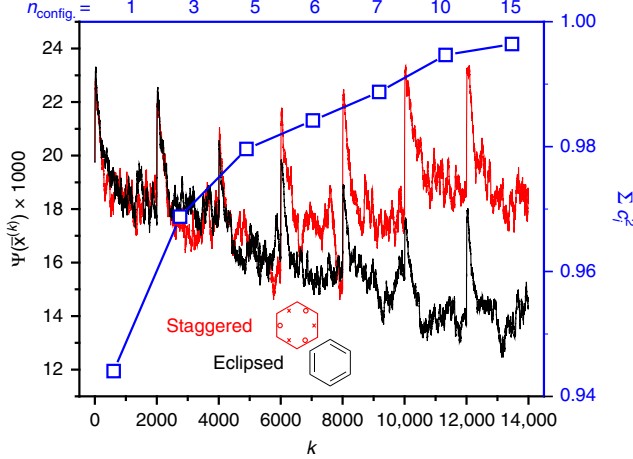

**Fig. 2 Analysis of the (30,18) CAS-CI wavefunction by DVMS.** In red and black, the wavefunction value at the Voronoi site for staggered and eclipsed Kekulé structures, as a function of $k$, the number of DVMS cycles. In blue, the cumulative completeness of the (30,18) CAS-CI wavefunction.

**Fig. 1 DVMS structures for benzene. a** Voronoi site for the RHF/6-31G(d) wavefunction. The electron positions of an arbitrary spin are shown as small yellow spheres. **b** Cross sections through the wavefunction around the Voronoi site in **a** C–C bonding electrons are shown as blue lobes. C–H bonds are shown in grey. **c**. Voronoi site showing staggered spins. The electron positions of each spin are respectively shown as small yellow and green spheres. **d**. Cross sections around the Voronoi site in **c**. The two spins of the C–C bonding electrons are shown in blue and red. C–H bonds are shown in grey.

Kekulé structure, with double bonds where single bonds are in Fig. 1a, b, and vice versa.

**Electron correlation**. At the single-determinant level of theory, for a given spin configuration the wavefunction is multiplicatively separable in each spin. As such, there is no preference for the spins to occupy the same or alternate Kekulé structures. We explored the effect of correlation by increasing the number of excited configurations in the wavefunction calculated using the CAS-CI approach. A CAS-CI wavefunction can be written in the form $\Psi = \sum_i c_i \psi_i$, where the $\psi_i$ represent different configurations. By truncating this sum, we can interrogate the effects of including successively more excitations on the overall wavefunction, and exploring the wavefunction by DVMS.

A valence (30,18) CAS-CI wavefunction was calculated from the RHF orbitals. While we only allowed occupancy of the lowest three unoccupied orbitals, we allowed excitation of all valence electrons to minimise the imposition of intuition.

The DVMS procedure was applied, with the wavefunction being modified every 2000 steps to include more configurations. Excited configurations were successively added in order of decreasing magnitude of the corresponding coefficient $c_i$. Degenerate configurations were added as a set, and the wavefunction amplitude at the Voronoi site is plotted in Fig. 2. For each wavefunction (differing from the previous by the inclusion of more configurations), the walkers restarted at the Voronoi point of the previous, less correlated, wavefunction, and initial steps are then biased to higher wavefunction values by the Metropolis algorithm. This results in the spikes after each addition. What is clear, is that after about seven configurations, the wavefunction at the staggered Voronoi site, where the two spins occupy alternate Kekulé structures (Fig. 1c), exceeds that at the eclipsed Voronoi site (Fig. 1a), and can be taken as more probable. At this point the wavefunction is nearly 99% complete, as calculated.

The result agrees in spirit with Linnett's non-paired spatial-orbital wavefunction[19], which was argued on energetic grounds. Indeed, the staggered structure results from excited configurations in the MO wavefunction pushing the spins apart to reduce the Coulomb repulsion energy, by suppressing the magnitude of the wavefunction where they are superposed. At the 10-configuration level, all configurations correspond to a redistribution of electrons within the $\pi$-orbitals. While intuitive, we take this as motivation to analyse an all-$\pi$ CAS-CI (6,6) wavefunction below.

The value of the wavefunction at the Voronoi site as plotted in Fig. 2 is only a crude measure of the preference for eclipsed or staggered Kekulé structures. However, the Metropolis algorithm samples the wavefunction according to the probability density $|\Psi|^2$, and thus we can assess the preference for whole tile locations in $3N$-dimensional space by performing a Monte Carlo integration. To this end, we defined four Voronoi sites corresponding to the two eclipsed and two staggered Kekulé structures, related by 60° rotations of either or both of the C–C bonding electron spin sets of the Voronoi site located above. Using the CAS-CI (6,6) wavefunction, a Metropolis sampling random walk of 500 walkers was performed, commencing at a single eclipsed structure, and the walkers were allocated to the appropriate Voronoi cell (staggered or eclipsed) on the fly.

The results are shown in Fig. 3. With seven configurations (Fig. 3a), the walkers distribute themselves between the staggered and eclipsed sites within 500 steps. The traces in the blocks 501–1000, 1001–1500 and 1501–2000 were not statistically different. The traces were fit to a kinetic model, with the rate constant for crossing from one cell to the other found to be about 0.005 step$^{-1}$. As such, it can be estimated that the walker positions are decorrelated after 200 steps, and so standard deviations in the Voronoi cell occupancy were divided by $\sqrt{n_{\text{step}}/200}$ to determine the standard error in the mean. The walkers display a slight preference for the staggered Voronoi sites. The occupancies of the eclipsed and staggered forms are $241 \pm 6$ and $259 \pm 6$ ($2\sigma$ error).

With 11 configurations, we see a largely unchanged preference for the staggered Kekulé over the eclipsed Kekulé forms. Out of 500 walkers $260 \pm 7$ favour the staggered form and $240 \pm 7$ remain in one of the eclipsed forms ($2\sigma$ error). At 21 configurations, the result is even clearer: $267 \pm 8$ favour the staggered form and $233 \pm 8$ remain in one of the eclipsed forms ($2\sigma$ error). The equilibrated populations are shown as a bar graph in Fig. 4.

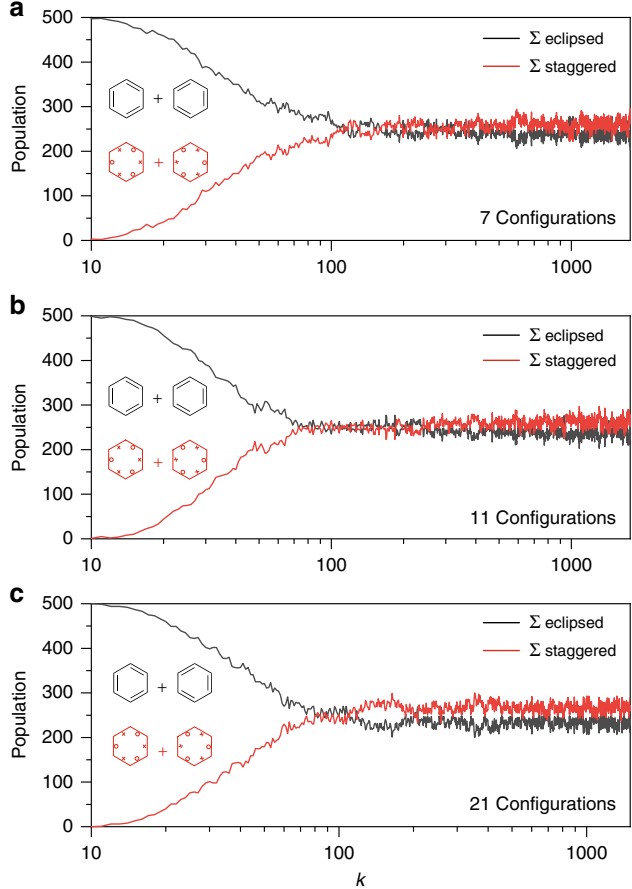

**Fig. 3 Analysis of the (6,6) CAS-CI wavefunction by Metropolis sampling.** The populations in the Voronoi cells, corresponding to eclipsed (black) and staggered (red) arrangements of the C–C bonding electrons, are plotted for three truncations of the (6,6) CAS-CI wavefunction sum, as a function of the Monte Carlo step number, $k$: **a** 7 configurations, **b** 11 configurations, **c** 21 configurations.

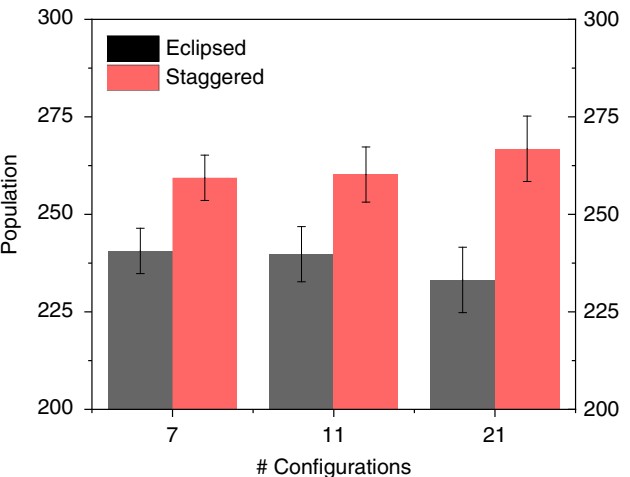

**Fig. 4 The equilibrium populations of eclipsed and staggered Kekulé structures ($2\sigma$ error).** As the number of configurations contributing to the wavefunction is increased, there emerges a distinct preference for the staggered Kekulé structures over the eclipsed Kekulé structures.

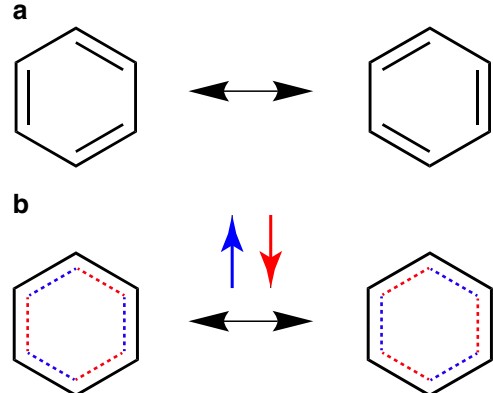

**Fig. 5 Diagrammatical depictions of resonance in benzene. a** The classical Kekulé structures. **b** The description suggested here, where different spins are coloured red and blue.

## Discussion

The pairing of electrons is an idea that pervades chemistry. It is ultimately rooted in the bonding description of Lewis and the filling of MOs according to the Aufbau principle. In an electron pair there is always one of each spin. However, electron correlation cannot be described by a single electron configuration, and electrons of differing spin repel one another by the same Coulomb force as those of the same spin. As such, while electrons of the same spin avoid one another by virtue of the structure of the wavefunction, electrons of differing spin do so through either wholesale rearrangement (static correlation) or detailed avoidance (dynamic correlation).

In a molecule such as water, the Pauli repulsion of like-spin electrons determines the tetrahedral structure which must be the same for each spin because two vertices are pinned to the O–H bonds[25,31]. As such, there is no static correlation. However, as shown for $C_2$[25,30], the effect of electron correlation is to arrange the eight valence electrons into two spin sets with each arranged tetrahedrally, but with the tetrahedral pointing in different directions. This leaves three electrons of each spin in the bonding region and one of each spin anti-correlated outside the bonding region on the C–C axis. This is difficult to visualise by inspection of a configuration interaction vector, which is a series of numbers corresponding to coefficients of the various configurations giving rise to the overall wavefunction. The wavefunction tiling method overcomes these difficulties by inspecting the occupancy of tiles which can be directly related to Lewis-structures, albeit without necessarily pairing the electrons.

It is by allowing the electrons to un-pair that the effects of correlation are seen to manifest. As shown in benzene, there is a preference for staggered Kekulé structures where the electron density is evenly distributed between the C–C bonds. In the staggered form there are three electrons in each C–C bonding region but in the eclipsed forms there are three bonds with four electrons and three bonds with two (Fig. 5).

Wavefunction tiles have proven themselves to be of great utility in extracting chemically intuitive motifs from MO wavefunctions. They recover lone-pairs, banana-bonds and curly arrow mechanisms from rigorous calculations. Here, we have shown that they also allow the inspection of the effects of electron correlation. While a correlated wavefunction in MO theory is described as the sum of a ground and many excited configurations with various coefficients, we have shown that it may be visualised in terms of wavefunction tiles to discern the electronic structure and how this differs from the uncorrelated, single-determinant wavefunction.

In benzene, wavefunction tiles demonstrate that Kekulé structures lurk within the anti-symmetrised MO wavefunction. However, it is by unpairing the spins and allowing the occupancy of staggered Kekulé structures that electron correlation manifests.

With 21 configurations it was demonstrated that the structure of benzene is 53% staggered. This underlines that a complete qualitative description of electronic structure, where resonance is exhibited, requires the unpairing of spins.

## Methods

**Electronic wavefunctions**. A single-determinant wavefunction of benzene was obtained at the RHF/6-31G(d)//B3LYP/6-31G(d) level of theory, using the GAMESS programme[32]. The correlated wavefunction was obtained at the CAS-CI level of theory using the Hartree–Fock orbitals.

**Dynamic voronoi metropolis sampling**. The DVMS procedure explores the calculated $N$-electron wavefunction with a swarm of walkers which each commencing at the same point in the $3N$-dimensional space, typically a local wavefunction maximum. Walkers take random Gaussian-distributed steps with variance 0.2 $a_0$ in each dimension. The step is accepted according to the Metropolis probability $p = \left|\Psi(\mathbf{x}^{(k+1)})\right|^2 / \left|\Psi(\mathbf{x}^{(k)})\right|^2$, where $k$ is the step number. Steps are always rejected if they result in a sign-change (node-crossing). If walkers leave the Voronoi cell defined by the average walker position, $\bar{\mathbf{x}}$, and the set of even-signed permutations thereof, the permutation of electrons is found to return the walker to the original cell. As the average walker position changes, so too does the dynamic Voronoi diagram. The procedure is iterated until convergence is achieved.

## Data availability
The data that support the findings of this study are available from the corresponding author upon reasonable request.

## Code availability
The DVMS code is available from the authors upon request.

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

## Acknowledgements
This work was supported by the Australian Research Council (Centre of Excellence in Exciton Science CE170100026).

## Author contributions
Y.L. performed the calculations and wrote computer code. P.K. designed the algorithm to solve the restricted assignment problem. T.W.S. conceived the method. T.J.F. and T.W.S. designed the study and wrote the manuscript.

## Competing interests
The authors declare no competing interests.
