## [Peer Review File · Nature Communications]

Reviewers' comments:

Reviewer #1 (Remarks to the Author):

This paper by Liu and co-workers describes a novel and creative approach for visualizing electronic structure with an application to benzene. The approach is one that provides a new, wave-function-based method for understanding the most probable location of electrons and provides physical underpinnings of bonding. This contribution focuses on the bonding in benzene – a classic problem in chemical bonding.

Overall, I found this work to be of broad and general interest to the chemistry community, and with the level of impact that I would expect for work published in Nature Communications. Based on this, I recommend publication once the authors have had the opportunity to consider the following comments:

End of second paragraph on page 3: “the one-electron orbitals” should be “the one-electron spin-orbitals”

Start of the third paragraph on page 3: “The sign of the wave function must change” should be “the sign of the spatial part of the wave function must change”

End of the third paragraph on page 3 – the statement that Hartree-Fock theory does not include antisymmetry is incorrect. The original Hartree theory, which is what is discussed by Slater, did not include antisymmetry. By introducing the Fock operator, antisymmetry is considered in the development of the direct product form of the wave function.

Middle of page 4: N_α and N_β have not been defined, and I think the permutation should be not on all electrons (end of sentence) but only those with the same spin.

End of the 2nd paragraph on page 4: it is not clear what “self-consistent” means in this context.

Page 6, start of the 1st paragraph – it’s not clear what the authors are referring to by “cross-terms in the wave function.”

Page 6, middle of the third paragraph – “ensemble” is an unusual word choice I think the authors mean “as a group” since ensemble takes on a different meaning within the context of Monte Carlo sampling also used in this study.

First line of the caption to Figure 2: This sentence is confusing, please rewrite.

Page 9, end of second paragraph: Metropolis should be in upper case. Later in the paragraph, “depermutation” is not a clear description of what is done. A more specific description would be helpful.

Figure 3: The caption could be clearer. I believe that the two top and two lower panels show the same data on different scales, although that is not clear from the caption.

Page 13, end of discussion: The comment about calculations on allyl and ozone. If the authors would like to include this work, it should be discussed in greater detail. If not, and it’s not already published, the work should be reported completely in a future publication.

Finally, the comment on data availability doesn’t seem to meet the standards of public availability of data that may journals currently expect. It would be preferable if the authors could make their data available either through a publicly accessible data repository or as part of the supporting information for this paper. How the authors should address this final point is an issue of editorial

policy and I defer to the editor on this point.

Reviewer #2 (Remarks to the Author):

The electronic structure of benzene is well understood as is the equivalence of VB and canonical MO-type descriptions. Nonetheless, both methods are often overinterpreted, and it is of substantial interest to discuss the analysis of the full 3n-dimensional wavefunction, be it based on canonical, localized, or atomic orbitals. In this paper, the application of the DVMS method of the authors is applied to benzene. VB-like Kekulé structures are obtained from MO-type wavefunctions. Most interestingly, a structure with 6 equivalent 3 electron bonds is obtained additionally. Both are chemically intuitive, yet obtained rigorously from computation. The general description does not depend on the details of the wavefunction, but with increasing correlation the 3 electron bond description becomes slightly preferred.

The conclusions are original and well justified. The paper is recommended for publication after minor corrections:

(1) Fig. 2 requires a little more explanation: k is not explained (probably the Metropolis step number)

(2) Are the blue and black axis really correctly scaled? Shouldn't the "spikes" coincide with the addition of new determinants?

(3) Last paragraph p6: The explanation of the "spikes" is unclear. If all walkers are put at the Voronoi centroids then a near maximal value for Psi is expected. Metropolis steps would then explore also points with somewhat lower probability. That would explain the spike pattern. The authors write that the initial steps are "biased to higher wavefunction values by the Metropolis algorithm". Then the black/red curve should increase after adding determinants.

(4) p11 last paragraph: "static correlation". Because of the dominance of the first determinant one should not speak of "static correlation", but of dynamic correlation which is the avoidance of opposite spin electrons explaining here well the preference.

typos: p1 heirarchy, p9 Voronoi side

Reviewer #3 (Remarks to the Author):

I think the paper presents a possibly interesting analysis / picture of electron correlation. I think the details can be presented in a clearer fashion however, and I think a closer analysis also raises questions.

I think it would be good to discuss the underlying CASSCF wave function in some more detail: (30,18) means 30 electrons in 18 spatial orbitals, or 15 alpha and 15 beta electrons. I think the authors could clarify this in regards to figure 1 (# of points), and perhaps use another colour for points that denote both alpha and beta electrons (or an electron pair). Of these 12 electron/ 6 orbitals refer to the less interesting CH bonds. Left are 18 electrons in 12 orbitals. 12 e- are conventionally referred to as sigma electrons (12 e- / 6 orbitals) while we have 6 pi-electrons in $3+3 = 6$ bonding /anti-bonding pi orbitals.

I think the CASSCF wave function might give a reasonable description of electron correlation in the pi system. The correlation in the sigma system is dubious, as the corresponding sigma* orbitals are not included in the CAS. Hence, I think the qualitative picture of electron correlation may be significantly distorted (I don't know). Because of this I have little confidence in any of the detailed analysis (53% staggered etc.).

I think it could be interesting to consider results for the smaller (6,6) calculation in the pi system. Does this lead to staggered and eclipsed configurations? It appears such configurations are not

"orthogonal", and I am not entirely sure what it means or how stable the results can be reproduced (initial guesses). On further thought non-orthogonal configurations are widely used in a VB context, and this seems fine (the authors may add to manuscript). My naive hunch is that for the CAS(6,6) the result might be 50-50 mostly by (not well thought out) symmetry arguments. The (artificial) sigma - pi* correlation may (slightly) perturb this picture, giving rise to the results presented in the paper.

An alternative very large CAS would include 12 more electrons and 12 spatial orbitals (sigma, sigma*) . That is not very well defined, as sigma* orbitals are not easily identified.

Given my reservations regarding these details, which may change the qualitative picture, I think the paper might not be suitable for nature communications. It may also be that the questions raised here can be addressed, while retaining the reported picture of electron correlation, leading to resubmission of the paper.

Marcel Nooijen

We reproduce the Reviewers' comments below, with our replies in **bold**.

Reviewer #1 (Remarks to the Author):

This paper by Liu and co-workers describes a novel and creative approach for visualizing electronic structure with an application to benzene. The approach is one that provides a new, wave-function-based method for understanding the most probable location of electrons and provides physical underpinnings of bonding. This contribution focuses on the bonding in benzene - a classic problem in chemical bonding.

Overall, I found this work to be of broad and general interest to the chemistry community, and with the level of impact that I would expect for work published in Nature Communications. Based on this, I recommend publication once the authors have had the opportunity to consider the following comments:

We thank the Reviewer for the positive comments.

End of second paragraph on page 3: "the one-electron orbitals" should be "the one-electron spin-orbitals"

Done.

Start of the third paragraph on page 3: "The sign of the wave function must change" should be "the sign of the spatial part of the wave function must change"

True. Done.

End of the third paragraph on page 3 - the statement that Hartree-Fock theory does not include antisymmetry is incorrect. The original Hartree theory, which is what is discussed by Slater, did not include antisymmetry. By introducing the Fock operator, antisymmetry is considered in the development of the direct product form of the wave function.

We had written "Furthermore, even if one were to insist upon employing the canonical MOs which result from Hartree-Fock theory, this approach altogether ignores antisymmetry: The Hartree-product is not a correct wavefunction." We think the referee means that because Hartree-Fock theory correctly accounts for antisymmetry that the use of these orbitals does not ignore it. The ignorance is in employing the orbitals without antisymmetry, which is the usual approach of chemists. We offer to clarify that we are talking about orbitals coming from correctly antisymmetrised Hartree-Fock theory and to replace *this approach altogether* with *interpretations that view these orbitals individually*.

Middle of page 4: N_α and N_β have not been defined, and I think the permutation should be not on all electrons (end of sentence) but only those with the same spin.

We have added the definition and added the phrase "of the same spin".

End of the 2nd paragraph on page 4: it is not clear what "self-consistent" means in this context.

We have added the phrase "of the tile that it defines" as well as the clarifying sentence "Self-consistency is required as \bar{x} depends on R , while R simultaneously depends on \bar{x} ."

Page 6, start of the 1st paragraph - it's not clear what the authors are referring to by "cross-terms in the wave function."

We have rephrased this "At the single-determinant level of theory, for a given spin-configuration the wavefunction is multiplicatively separable in each spin"

Page 6, middle of the third paragraph - "ensemble" is an unusual word choice I think the authors mean "as a group" since ensemble takes on a different meaning within the context of Monte Carlo sampling also used in this study.

This has been changed to "set".

First line of the caption to Figure 2: This sentence is confusing, please rewrite.

The first sentence of the caption has been replaced with "In red and black, the wavefunction value at the Voronoi site for staggered and eclipsed Kekulé structures, as a function of k , the number of DVMS cycles. The wavefunction is modified every 2000 steps to include more configurations, added in order of the magnitude of the coefficients of the configurations in the CI expansion." Note that previously the colours were specified incorrectly.

Page 9, end of second paragraph: Metropolis should be in upper case.

Done.

Later in the paragraph, "depermutation" is not a clear description of what is done. A more specific description would be helpful.

We have rephrased this to "and the walkers were allocated to the appropriate Voronoi cell (staggered or eclipsed) on the fly." The depermutation is not really important but is our way of keeping track of the walkers.

Figure 3: The caption could be clearer. I believe that the two top and two lower panels show the same data on different scales, although that is not clear from the caption.

In the original figure, the top and bottom are different numbers of configurations and the left and right are different time scales, but on the right the data is summed, as indicated.

In order to reduce confusion and also in response to Referee #3, we have redrawn Figure 3 with only the summed data for the new results on the (6,6) wavefunction suggested.

We have updated the caption and added guides to the chosen colour key to help the reader.

Page 13, end of discussion: The comment about calculations on allyl and ozone. If the authors would like to include this work, it should be discussed in greater detail. If not, and it's not already published, the work should be reported completely in a future publication.

We have removed these two sentences.

Finally, the comment on data availability doesn't seem to meet the standards of public availability of data that may journals currently expect. It would be preferable if the authors could make their data available either through a publicly accessible data repository or as part of the supporting information for this paper. How the authors should address this final point is an issue of editorial policy and I defer to the editor on this point.

The wording was suggested by Nature Chemistry. The detailed data cannot be reproduced without the same random number generator, so isn't useful. That's the nature of Monte Carlo techniques. The aggregated data is reported in the manuscript and we are happy to share anything reasonably requested.

Reviewer #2 (Remarks to the Author):

The electronic structure of benzene is well understood as is the equivalence of VB and canonical MO-type descriptions. Nonetheless, both methods are often overinterpreted, and it is of substantial interest to discuss the analysis of the full 3n-dimensional wavefunction, be it based on canonical, localized, or atomic orbitals. In this paper, the application of the DVMS method of the authors is applied to benzene. VB-like Kekulé structures are obtained from MO-type wavefunctions. Most interestingly, a structure with 6 equivalent 3 electron bonds is obtained additionally. Both are chemically intuitive, yet obtained from rigorously from computation. The general description does not depend on the details of the wavefunction, but with increasing correlation the 3 electron bond description becomes slightly preferred.

The conclusions are original and well justified. The paper is recommended for publication after minor corrections:

We thank the Reviewer for the positive comments.

- (1) Fig. 2 requires a little more explanation: k is not explained (probably the Metropolis step number)

Correct. The modified caption indicates that data is shown "as a function of k , the number of DVMS cycles."

- (2) Are the blue and black axis really correctly scaled? Shouldn't the "spikes" coincide with the addition of new determinants?

The axes are correctly scaled. What happens is that the walkers are all returned to the Voronoi site after the addition of more configurations. The next few steps are all biased to higher wavefunction value, which causes the spike. This is explained in the manuscript as "For each wavefunction, the walkers started at the Voronoi point of the previous equilibration, and initial steps are then biased to higher wavefunction values by the Metropolis algorithm. This results in the "spikes" after each addition." Note that the points indicating the sum of squares of the coefficients for each number of configurations (the blue data) do not coincide with the change in the wavefunction.

(3) Last paragraph p6: The explanation of the "spikes" is unclear. If all walkers are put at the Voronoi centroids then a near maximal value for Psi is expected.

This is not true. Recall that the wavefunction magnitude shown is that of the Voronoi site, not the sum or average of the values at the walkers. Note also that there is a big difference between the maximum of the wavefunction and the equilibrated DVMS structure. This is explained in our PCCP paper.

Metropolis steps would then explore also points with somewhat lower probability. That would explain the spike pattern. The authors write that the initial steps are "biased to higher wavefunction values by the Metropolis algorithm". Then the black/red curve should increase after adding determinants.

They do. The wavefunction value increases for the first few steps after the addition of configurations, for the reasons outlined. After the walkers spread out the Voronoi site is "dragged" to regions of even lower wavefunction magnitudes.

(4) p11 last paragraph: "static correlation". Because of the dominance of the first determinant one should not speak of "static correlation", but of dynamic correlation which is the avoidance of opposite spin electrons explaining here well the preference.

We have removed the word "static" from that instance.

typos: p1 heirarchy, p9 Voronoi side

Thanks.

Reviewer #3 (Remarks to the Author):

I think the paper presents a possibly interesting analysis / picture of electron correlation. I think the details can be presented in a clearer fashion however, and I think a closer analysis also raises questions.

I think it would be good to discuss the underlying CASSCF wave function in some more detail:

It is a CAS-CI wavefunction, not CASSCF. The orbitals used are from Hartree-Fock. This has now been made clearer.

(30,18) means 30 electrons in 18 spatial orbitals, or 15 alpha and 15 beta electrons. I think the authors could clarify this in regards to figure 1 (# of points), and perhaps use another colour for points that denote both alpha and beta electrons (or an electron pair).

We have added the phrase “of an arbitrary spin” to the description of panel a. Note that the depictions of the equilibrated Voronoi site like those given in Figure 1 depict *all N* electrons of the given spin, not just those that may have been affected by the correlation treatment in the electronic structure theory calculation.

Of these 12 electron/ 6 orbitals refer to the less interesting CH bonds. Left are 18 electrons in 12 orbitals. 12 e- are conventionally referred to as sigma electrons (12 e- / 6 orbitals) while we have 6 pi-electrons in 3+3 = 6 bonding /anti-bonding pi orbitals.

I think the CASSCF wave function might give a reasonable description of electron correlation in the pi system.

We agree (CAS-CI wavefunction).

The correlation in the sigma system is dubious, as the corresponding sigma* orbitals are not included in the CAS. Hence, I think the qualitative picture of electron correlation may be significantly distorted (I don't know). Because of this I have little confidence in any of the detailed analysis (53% staggered etc.).

We appreciate this comment and have performed additional calculations to address this and the further comments below. We found that sigma orbitals were only involved at the 15-configuration level, which hardly altered the observed correlation. As such, we use this as motivation to perform the (6,6) calculation which is intuitive. However, we try to be careful to avoid the imposition of intuition in this study.

I think it could be interesting to consider results for the smaller (6,6) calculation in the pi system. Does this lead to staggered and eclipsed configurations?

We have performed an all-pi (6,6) calculation and now analyse this instead of the (30,18) wavefunction, which is only invoked for Figure 2. We extend the number of configurations to 21 in this all-pi space and are happy to report that not only is the effect robust, but the magnitude of the effect is robust. We are thus extremely confident in our findings. These are reported in the revised Figure 3 and 4.

It appears such configurations are not "orthogonal", and I am not entirely sure what it means or how stable the results can be reproduced (initial guesses). On further thought non-orthogonal configurations are widely used in a VB context, and this seems fine (the authors may add to manuscript).

We understand that the Reviewer refers to non-orthogonal orbitals being used in VB calculations. Our spatial configurations (as opposed to orbital occupancies) are projections of 126-dimensional vectors and are not related to orbitals, though the tile isosurfaces resemble them. We do not feel that making this connection will be beneficial.

My naive hunch is that for the CAS(6,6) the result might be 50-50 mostly by (not well thought out) symmetry arguments.

As shown in the revised manuscript, the CAS(6,6) wavefunction recovers the correlation originally calculated, because it contains the same configurations with largely similar coefficients to that originally calculated.

The (artificial) sigma - pi* correlation may (slightly) perturb this picture, giving rise to the results presented in the paper.

The only sigma-pi* correlation was at the original 15 configuration level which did not deviate significantly from the 10-configuration level.

An alternative very large CAS would include 12 more electrons and 12 spatial orbitals (sigma, sigma*) . That is not very well defined, as sigma* orbitals are not easily identified.

We agree that the sigma* orbitals are not easily identified. The suggestion of a (6,6) CAS was a good one, and makes the paper much tighter and intuitively sound.

Given my reservations regarding these details, which may change the qualitative picture, I think the paper might not be suitable for nature communications. It may also be that the questions raised here can be addressed, while retaining the reported picture of electron correlation, leading to resubmission of the paper.

We sincerely appreciate these comments which have forced us to tighten up the paper and focus on the (6,6) wavefunction. As expected (because the configurations chosen in the original wavefunction were largely the same), the quantitative results were found to be robust.

Marcel Nooijen

We appreciate Marcel's comments.

REVIEWERS' COMMENTS:

Reviewer #1 (Remarks to the Author):

The authors fully addressed my comments and I believe that the paper is now suitable for publication

Reviewer #3 (Remarks to the Author):

The authors gave clear responses to questions by all the referees. I think this paper is quite interesting and I am convinced now of the validity of the analysis. I have only some minor remaining comments that can presumably be addressed easily.

1. The sentence: "But, interpreting the electronic structure of a molecule in terms of the one-electron spin-orbitals generated by MO theory has several major and irreparable flaws. " leaves me puzzled. I think the authors should say 'However, interpreting ...' I think the authors then go on in the next paragraph to explain what are (some) flaws. This is hard to interpret/understand. Keep these sections together in one paragraph, or use citation with the first sentence?

2. Still in the introduction:

" In this work, we revisit the electronic structure of benzene using the DVMS procedure. It is found that a single determinant wavefunction reproduces Kekulé structures, but with the two spins sets ambivalent to the other's occupation of one Kekulé structure or the other. "

I am unclear what 'ambivalent' means here. It is not precise. I think the authors might mean 'agnostic'. The probabilities for the Kekulé structure are completely (exactly) uncorrelated.

3.

The electrons of each spin alternately single- or double-bond between carbons, in accord with the Kekulé structure(s).

Missing "occupy" or something?

4.

At the single-determinant level of theory, for a given spin-configuration the wavefunction is multiplicatively separable in each spin. As such, whether the spins occupy the same or alternate Kekulé structures cannot be determined – they are equivalent.

I find this a confusing statement. I would think: the DVMS method is independent of method used. It works always, for any antisymmetric wave function (i.e. I object to: cannot be determined??) . For a Hartree-Fock wave function for benzene it yields staggered and eclipsed Kekulé structures with equal weight. This is a consequence of the closed-shell single determinant structure (I think ...). The structure is like $(A+B)_{\alpha} * (A+B)_{\beta}$. "structures" A and B are related by symmetry, the (spin-like) product structure arises from closed shell arguments -> equal weight (a mathematical proof, cast in stone).

These are all minor comments, easily corrected in a (final) revision.

REVIEWERS' COMMENTS:

Reviewer #1 (Remarks to the Author):

The authors fully addressed my comments and I believe that the paper is now suitable for publication

We thank the reviewer for their time and effort.

Reviewer #3 (Remarks to the Author):

The authors gave clear responses to questions by all the referees. I think this paper is quite interesting and I am convinced now of the validity of the analysis. I have only some minor remaining comments that can presumably be addressed easily.

We thank the reviewer for their time and effort in improving our manuscript.

1. The sentence: "But, interpreting the electronic structure of a molecule in terms of the one-electron spin-orbitals generated by MO theory has several major and irreparable flaws. " leaves me puzzled. I think the authors should say 'However, interpreting ...' I think the authors then go on in the next paragraph to explain what are (some) flaws. This is hard to interpret/understand. Keep these sections together in one paragraph, or use citation with the first sentence?

We remain convinced that the best way to start that sentence is with the word "but". This is more forceful than "however", but the meaning is largely similar (https://wordcounter.net/blog/2016/10/26/102560_can-you-start-a-sentence-with-but.html). Similarly, that sentence is intended to lead into the next paragraph, as is standard in academic writing. <https://waylink-english.co.uk/academic-writing/linking-paragraphs> But, to appease the reviewer, we have moved the sentence to the next paragraph, where it can reference the previous one.

2. Still in the introduction: "In this work, we revisit the electronic structure of benzene using the DVMS procedure. It is found that a single determinant wavefunction reproduces Kekulé structures, but with the two spins sets ambivalent to the other's occupation of one Kekulé structure or the other. "

I am unclear what 'ambivalent' means here. It is not precise. I think the authors might mean 'agnostic'. The probabilities for the Kekulé structure are completely (exactly) uncorrelated.

Both ambivalent and agnostic are anthropomorphising. We have already used the word agnostic in a different sense. We offer the word "equivocal".

3.

The electrons of each spin alternately single- or double-bond between carbons, in accord with the Kekulé

structure(s).

Missing "occupy" or something?

Here "bond" is used as a verb. So technically single and double should take adverb forms "singly" and "doubly", which it has now been amended to.

4. At the single-determinant level of theory, for a given spin-configuration the wavefunction is multiplicatively separable in each spin. As such, whether the spins occupy the same or alternate Kekulé structures cannot be determined - they are equivalent.

I find this a confusing statement. I would think: the DVMS method is independent of method used. It works always, for any antisymmetric wave function (i.e. I object to: cannot be determined??) . For a Hartree-Fock wave function for benzene it yields staggered and eclipsed kekule structures with equal weight. This is a consequence of the closed-shell single determinant structure (I think ...). The structure is like $(A+B)_\alpha * (A+B)_\beta$. "structures" A and B are related by symmetry, the (spin-like) product structure arises from closed shell arguments -> equal weight (a mathematical proof, cast in stone).

We suggest "As such, there is no preference for the spins to occupy the same or alternate Kekulé structures."

These are all minor comments, easily corrected in a (final) revision.

We again thank the reviewer for their time and effort.